# Characterization of ^18^F-PM-PBB3 (^18^F-APN-1607) Uptake in the rTg4510 Mouse Model of Tauopathy

**DOI:** 10.3390/molecules25071750

**Published:** 2020-04-10

**Authors:** Chi-Chang Weng, Ing-Tsung Hsiao, Qing-Fang Yang, Cheng-Hsiang Yao, Chin-Yin Tai, Meng-Fang Wu, Tzu-Chen Yen, Ming-Kuei Jang, Kun-Ju Lin

**Affiliations:** 1HARC and Department of Medical Imaging and Radiological Sciences, Chang Gung University, Taoyuan 333, Taiwan; ccweng@mail.cgu.edu.tw (C.C.-W.); ihsiao@mail.cgu.edu.tw (I.-T.H.); forchem12@gmail.com (Q.-F.Y.); 2Department of Nuclear Medicine and Center for Advanced Molecular Imaging and Translation, Linkou Chang Gung Memorial Hospital, Taoyuan 333, Taiwan; 3APRINOIA Therapeutics Inc., Taipei 11503, Taiwan; cytai@aprinoia.com (C.-Y.T.); MFWu@aprinoia.com (M.-F.W.); yentc1110@gmail.com (T.-C.Y.); mkjang@aprinoia.com (M.-K.J.)

**Keywords:** ^18^F-FM-PBB3, ^18^F-APN-1607, Alzheimer’s disease, small animal PET, tauopathy, transgenic mice

## Abstract

Misfolding, aggregation, and cerebral accumulation of tau deposits are hallmark features of Alzheimer’s disease. Positron emission tomography study of tau can facilitate the development of anti-tau treatment. Here, we investigated a novel tau tracer ^18^F-PM-PBB3 (^18^F-APN-1607) in a mouse model of tauopathy. Dynamic PET scans were collected in groups of rTg4510 transgenic mice at 2–11 months of age. Associations between distribution volume ratios (DVR) and standardized uptake value ratios (SUVR) with cerebellum reference were used to determine the optimal scanning time and uptake pattern for each age. Immunohistochemistry staining of neurofibrillary tangles and autoradiography study was performed for ex vivo validation. An SUVR 40–70 min was most consistently correlated with DVR and was used in further analyses. Significant increased ^18^F-PM-PBB3 uptake in the brain cortex was found in six-month-old mice (+28.9%, *p* < 0.05), and increased further in the nine-month-old group (+38.8%, *p* < 0.01). The trend of increased SUVR value remained evident in the hippocampus and striatum regions except for cortex where uptake becomes slightly reduced in 11-month-old animals (+37.3%, *p* < 0.05). Radioactivity distributions from autoradiography correlate well to the presence of human tau (HT7 antibody) and hyperphosphorylated tau (antibody AT8) from the immunohistochemistry study of the adjacent brain sections. These findings supported that the 40–70 min ^18^F-PM-PBB3 PET scan with SUVR measurement can detect significantly increased tau deposits in a living rTg4510 transgenic mouse models as early as six-months-old. The result exhibited promising dynamic imaging capability of this novel tau tracer, and the above image characteristics should be considered in the design of longitudinal preclinical tau image studies.

## 1. Introduction

Alzheimer’s disease (AD) is the most common neurodegenerative disease in the world [1]. More than one percent of the population over 65-years-old worldwide is assumed to be affected by AD this decade. The pathogenesis mechanism of this disease so far is not clear but the two hallmarks of the disease, beta-amyloid plaques [2] and phosphorylated tau protein induced neurofibrillary tangles [3], may help us to find possible clues for the disease-causing mechanism.

Beta-amyloid plaques and neurofibrillary tangles are assumed to play a major role in the death of memory-formation-related neurons and inducing the clinical symptoms of dementia for the AD patients in the end. To uncover the mystery of how these devastating proteins work in a living AD patient’s brain, positron emission tomography (PET) combing specific radiotracers to target beta-amyloid plaques or aberrant tau proteins provides a chance to study the pathogenesis mechanism of AD [4]. In the past two decades, several PET ligands have been proven to efficiently bind to the plaques and some have even been approved by the FDA, such as Amyvid,^18^F-flutemetamol, and ^18^F-florbetaben [5,6,7,8]. Though this kind of beta-amyloid plaque targeting ligands have shown great success via in vivo binding characteristics, the binding strength seems to show only slight association with the clinical symptom outcomes [9], which raises the question as to whether these plaques are more likely to affect the process or results of AD rather than the pathogen of the disease.

Recently, highly phosphorylated tau proteins have been thought to be another possible cause of AD progression due to the validation of their relationship with the clinical severity of dementia [4]. For such reasons, many tau protein specific radiotracers were developed for imaging the abnormal tau protein burden in the AD patients or even on the transgenic animal models [10,11]. ^18^F-THK5317 has provided the first evidence of the correlation between the powers of the in vivo imaging of PET with the symptom performance of patients [12,13]. However, this first generation of a tau-specific PET tracer was abandoned due to the relatively high level of nonspecific binding in white matter, possibly binding to the myelin basic proteins. Afterward, another advanced radioligand, ^18^F-THK5351, was developed with much lower white matter binding and higher binding specificity only focusing on the abnormal tau-associated regions [14,15]. However, the dominant binding was delineated from ^18^F-THK5351 in the brain regions such as the basal ganglia, thalamus, and brainstem and this tracer’s off-target binding to monoamine oxidase-B (MAO-B) has been verified [16,17]. Like ^18^F-THK5351, ^18^F-Flotaucipir has also demonstrated superior binding properties specific to tau proteins [18,19,20]; however, this tracer also has the off-targeting issue, and it has been further confirmed that this non-specific binding could be derived from monoamine oxidase-A (MAO-A) or MAO-B [21,22].

Maruyama and colleagues published another tau-specific radiotracer, ^11^C-PBB3, in 2013, which has been evaluated and has shown a high specific binding affinity to phosphorylated tau proteins [23]. Though this tracer not only reveals a roughly 30–50-fold higher tau-specific binding affinity compared to beta-amyloid plaques, its retention in the medial temporal region, the precuneus, and the frontal cortex, shows good correlation to the cognitive decline in AD patients [24,25]. However, the short half-life, low dynamic range, and metabolic instability limit the application of this tracer in AD research [4]. To solve the problems mainly coming from the half-life issue of this tracer, an F-18 radiolabeled ^18^F-PM-PBB3 (^18^F-APN-1607)/1-(fluoro-18F)-3-((2-((1E,3E)-4-(6-(methylamino)pyridin-3-yl)buta-1,3-dien-1-yl)benzo[d]thiazol-6-yl)oxy) propan-2-ol was synthesized and has also shown comparable binding characteristics to ^11^C-PBB3 in vitro (K_d_ = 9.9 nM for ^18^F-PM-PBB3 and K_d_ = 6.3 nM for ^11^C-PBB3 on the AD frontal cortex) [24,26].

Recently, Huang et al. investigated the optimal scanning time window of this novel tracer [27] and showed promising results for detecting tau distribution in both AD and non-AD subjects in a phase 0 study (manuscript under review). Currently, the strategy for developing a new treatment for neurodegenerative disorder relies partially on preclinical animal models. We propose the ability of non-invasive monitor tau aggregation using this novel radiotracer for addressing the bottleneck in drug discovery and development for dementia. For a better understanding of the strength of this novel tracer in detecting tau protein aggregation, different ages of transgenic mice with tauopathies, rTg4510 [28], were used in this study to evaluate the tracer sensitivity and aberrant tau progression in the different age groups based on this tracer.

## 2. Results

### 2.1. In Vivo Animal PET Studies

After the 1.5 h PET image acquisition, the time–activity curves (TACs) of the standardized uptake value ratio (SUVR) values from the different brain regions (cortex, hippocampus, midbrain, and striatum) were selected based on the Magnetic Resonance (MR) image template, as displayed in Figure 1. The tracer uptake of different brain regions reached the plateau roughly 30–40 min after ^18^F-PM-PBB3 administration and then retained the SUVR signals until the end of the whole imaging procedure. Compared to the 2-mo-old animal group, the tracer accumulation in the brain regions of the cortex, hippocampus, and striatum demonstrated an increasing trend versus age for the other groups.

Figure 2 shows representative rainbow color-coded animal PET images at different ages overlaid on a T2 MR template. The static PET images were summed from dynamic frames obtained between 40–70min post tracer injection. Sagittal plane and three coronal views in hippocampus regions were illustrated for comparison. ^18^F-PM-PBB3 uptake in SUVR showed no significant difference in the cortical region of 4-mo-old transgenic mice when compared to that of the 2-mo-old baseline group (+15.0%, not significant). The radioactivity uptake was elevated in 6-mo-old animal (+28.9%, *p* < 0.05), and increased further in 9-mo-old group (+38.8%, *p* < 0.01). The trend of increased SUVR value becomes steady or slightly reduced in 11-mo-old animal (+37.3%, *p* < 0.05). Of note, when compared to the overlaid MR template, the brain PET size was smaller at this age group as compared to those of the younger mice, suggesting cortical atrophy in the elder animal (Appendix A).

To further simplify and open a new window for the future tau imaging application of this tracer on the rTg4510 transgenic animal model, the animal PET images from the different groups were further processed and evaluated based on the comparison of the regional SUVR and distribution volume ratios (DVR) to define the optimal scanning time. The regional DVR and SUVR across dynamic image sets were compared using Pearson’s correlation. Figure 3 displays the r^2^ value and slope of the regression between the DVR and regional SUVR measured at each time window for all age groups. For all regions except the mid-brain, the r^2^ values are higher than 0.90 after a scanning time of 40 min. For the above two performances, considering the image quality, the scanning window of 40–70 min post-injection is suggested as the optimal scanning time window for future application of ^18^F-PM-PBB3 in animal tau imaging studies.

Using the optimal scanning time window determined in the previous analysis, regional SUVR at each volume of interest (VOI)was calculated within 40–70 min post-injection for each animal. Regional SUVR from each age group were compared for the 4 target VOIs. The relationships of regional SUVR across all age groups are displayed in Figure 4. The same as in TACs, SUVR seems to reach a plateau at the age of 8-mo-old for cortex (Figure 4A), but still increases to the age of 11-mo-old for other regions (Figure 4B–D). To further confirm the difference between the SUVR vs. age effect in different brain regions, the quantification data of the SUVR values are summarized in Table 1. The result demonstrates the significant difference between all brain regions except the midbrain when the animals are 6-mo-old. Interestingly, for the brain region of the striatum, the tracer uptake kept increasing with age; however, in the midbrain, the tracer only showed a significant difference for 11-mo-old animals, which could mean that the midbrain region is the last region affected by the hyperphosphorelated tau protein accumulation.

### 2.2. Ex Vivo Autoradiography

To confirm the ^18^F-PM-PBB3 uptake from animal PET in the earliest age group of the rTg4510 transgenic mice (see Table 1), an ex vivo autoradiography study was performed in a 6-mo-old animal. The ^18^F-PM-PBB3 uptake in the brain and the quantification results from cortex, hippocampus, striatum, midbrain, and cerebellar regions were listed in Figure 5A,B. The high-resolution autoradiography revealed a spatial distribution of radioactivity in the cortex, striatum, and hippocampus comparable to those in PET image findings (Figure 2). The quantification result from the autoradiography implied the cortex and hippocampus showed the most severe tau protein burden.

### 2.3. Immunofluorescence Staining

Transgenic rTg4510 mice express a repressible form of human tau containing the P301L mutation that shows extensive development of hyperphosphorylated tau and filamentous tau aggregates in the brain [29]. To establish whether radioactivity distribution co-localized to the brain regions express transgenic P301S-htau, adjacent brain sections from the above autoradiography study were immunolabeled for the presence of human tau (HT7 antibody) and hyperphosphorylated tau (antibody AT8). Immunofluorescent imaging (Figure 5C) showed prominent signals in the cortex, hippocampus, and striatum comparable to the autoradiography findings (Figure 5A). In particular, AT8 antibody recognizes the pathological tau species found in AD patients [30], whereas HT7 antibody binds to all tau species, including the normal and abnormal species. The finding supports that ^18^F-PM-PBB3 binds to human pathological tau expressed in the rTg4510 animal model.

## 3. Discussion

Tauopathy has been found to have a critical relationship with the clinical symptoms of patients with dementia [9], and thus represents a possible means for evaluating the severity/progression of clinical dementia in AD patients by using an in vivo PET imaging technique to examine tau pathology. To our knowledge, the data reported herein should be the first in vivo ^18^F-PM-PBB3 PET study on the rTg4510 transgene mouse model. We applied a 90-min dynamic scan for all animals of different ages in this study in search of the optimal imaging time window and the tauopathy depositions among different ages. The sensitivity and optimal imaging time window of ^18^F-PM-PBB3 for the static imaging procedure have been evaluated and these protocols can be applied in future studies related to new therapy methods or drug development.

According to the published literature about the tau pathology burden in the rTg4510 mouse model, four target regions, including the cortex, hippocampus, striatum, and midbrain, were chosen for the PET image analysis in this study [28]. From the 1.5 h dynamic scans, we found that the radiotracer uptakes were mainly located in the brain regions such as the cortex, hippocampus, and striatum, which is comparable with previous findings [23], further confirming the tau-specific binding ability of ^18^F-PM-PBB3. The steady-state for each brain region from the TAC was found roughly 40 min after the tracer injection, which also matches the results of studies using C-11 labeled PBB3 [17,31]. The tracer kinetic distribution on the rTg4510 mouse brain regions is slower than ^18^F-THK523, which reached the plateau roughly 20–30 min post tracer injection [11]. However, ^18^F-THK523 retention was significantly lower in gray matter than in white matter, suggesting insufficient binding affinity of this tracer to tau aggregates [32]. While ^18^F-PM-PBB3 washout fast from cerebellum reference region (Figure 1) makes it feasible for visual inspection of tau deposits in the brain cortex (Figure 2). Owing to its high binding affinity [26], the radioactivity uptake measured by SUVR may potentially differentiate the stage of tauopathy in each age group of the Tg4510 animals to facilitate translational research.

To relate the radioactivity uptake measured by PET to the underlying biochemical process, the utilization of mathematical models to describe tracer kinetics is necessary. These methods are generally derived based on the theory for conventional tracer kinetic models such as spectral analysis, compartmental models, and graphical methods in which the plasma input function may be replaced by a reference region devoid of the specific binding sites [33]. Thus, the DVR measured by Logan plot with cerebellum as reference [34] was used as a “gold standard” in our study. To further reduce the complexity of the PET study, we estimated the transient equilibrium time window when SUVR as a proxy for DVR values was established. Compared to DVR, SUVR from an optimal imaging duration can be applied to a wider population because of the much shorter scanning time; this could also increase the possible applications for other tauopathy research such as progressive supranuclear palsy and cortical basal degeneration [35,36]. The DVR derived from the Logan graphical analysis model with the reference region of the cerebellum and the SUVR results presented herein show that the SUVR ratios 40 to 70 min after tracer injection had the best correlation with the DVR values. Moreover, as for the correlation and slope with each brain region, except for the midbrain, the rest of the regions presented an excellent correlation (r^2^ > 0.9) and linearity (slope close to one) between DVR and SUVR. The imaging time window shows a slightly different time compared to the ^18^F-THK5117 (20–50 min), possibly due to the different species of the transgenic mouse model of tauopathy used in both studies [10]. This difference was even larger when compared to a human study showing 90–110 min post injection as an optimal scanning time window [27]. Since there is no simple allometric scaling factor to transform animal results into human, independent study design for each species and disease model is necessary for its application.

Based on the correlation study mentioned previously, the SUVR values (40 to 70 min post injection) obtained from different VOIs were used to evaluate the tau deposition at different ages. Although mildly increased tau signal was observed in 4-mo-old animals, a significant elevation can only be seen at the age of 6-mo-old in our study. Santacruz et al. (2005) found pretangles developing as early as 2.5-mo-old in the rTg4510 mouse brain using multiple immunohistochemistry antibodies targeting tau protein [28]. The delayed detection of tau deposition using animal PET possibly due to scanner sensitivity limitation or the tracer used in this study, ^18^F-PM-PBB3, which makes it feasible to see the tau pathology only when these aberrant proteins become mature tangles but not for tau oligomers [37]. The increased tau deposition observed from PET images behaves differently among brain regions. The cortical accumulation of tau radioactivity signal reached its plateau at the age of around 8-mo-old. This result is comparable with a study using ^11^C-PBB3, which found the tau pathology burden in the forebrain region reached the maximum at roughly the age of 7-mo-old [37]. Meanwhile, the tau tracer deposition seemed to increase continuously till 11-mo-old in the striatum and hippocampus subcortical regions. In contrast, there was no significantly increased uptake in the midbrain region until 11-mo-old when compare to 2-mo-old animals. These temporal differences in ^18^F-PM-PBB3 uptake pattern against age may be related to tau spreading topography and severe brain cortical atrophy in the elder animals which have been observed in other tauopathy models [38].

To further validate the evidence of the tau specific binding ability of the ^18^F-PM-PBB3 tracer, the 6-mo-old animal group, which showed the first significant difference in tau pathology deposition compared to the 2-mo-old group was used for the ex vivo autoradiography study and the tau-protein-specific immunofluorescence staining. From the image data shown in Figure 5, the ^18^F-PM-PBB3 autoradiography results demonstrated the most tracer accumulated in the brain regions such as the frontal cortex, hippocampus, and striatum, which fully matches the in vivo PET imaging data. Moreover, based on the immunofluorescence staining results from the two different antibodies, AT8 and HT7, the ex vivo ^18^F-PM-PBB3 autoradiographic images displayed a promising correlation with the staining results, and the staining images also agreed with previously published data [31,39].

One limitation of this study is that we did not investigate the radiometabolites. In brain study, ideally, no radiometabolites should exist, since they may show different kinetics and radioactivity distribution than the parent compound and complicate the quantification. As reported by Ono et al., they observed fewer radiometabolites of ^18^F-PM-PBB3 in mouse plasma and brain than those of PBB3 during the course after intravenous injection. Correspondingly, in our study, ex vivo autoradiography and immunofluorescence staining showed comparable signals from the adjacent brain sections without significant evidence of off-target binding of the radiotracer. Future research for biodistribution, radiometabolite quantification, and longitudinal studies should provide useful information for preclinical study design.

## 4. Materials and Methods 

### 4.1. Radiochemistry

The tau PET tracer ^18^F-PM-PBB3 (Figure 6A) was synthesized with a multi-purpose fluorination module (GE Healthcare, Chicago, USA). The module setup is shown in Figure 6B and the details are listed in the Appendix A. The starting activity was 77.72 ± 10.5 GBq and the final product was purified by HPLC at the end of synthesis. The overall radiochemical yield was 28.30% ± 5.84% (decay-corrected), satisfying all quality control criteria (*n* = 110). It was stable for up to 6 h with high radiochemical purity as 95.86% ± 1.90% (the HPLC gradient condition and the representative HPLC profile were listed in Appendix A and Appendix A) and specific activity of 218.13 ± 115.77 TBq/mmol.

### 4.2. Animals

The breeding pair of rTg4510 [28] with strain backgrounds of FVB and 129S6 was licensed from the Mayo Clinic and bred at the National Laboratory Animal Center (NLAC, Taipei Taiwan), an AAALAC-accredited facility. The bitransgenic features of the F1 offspring carrying both CaMK2a-tTA and TetO-MAPT*P301L were determined by PCR genotyping from 10-day-old mouse toes. Both female and male mice were used throughout the study. Animal care was in compliance with the Guide for the Care and Use of Laboratory Animals. All animals were delivered from NLAC to The Division of Laboratory Animal Resources at Chang Gung Memorial Hospital (CGMH, Taoyuan, Taiwan) a few days prior to the surgical procedure for habituation. This study was approved by the Institutional Animal Care and Use Committee (IACUC No. 2017103001) at CGMH.

### 4.3. In Vivo Animal PET Studies

The PET image acquisition was performed using a Siemens Inveon PET scanner (Siemens Medical Solutions, Knoxville, TN, USA), and the protocol details are listed in the Appendix A. Briefly, three rTg4510 mice of each animal group (2-, 4-, 6-, 8-, 9-, 11-mo-old) were used in this study for optimizing the scanning time window and investigating the abnormal tau protein deposition against age effect. A 90-min dynamic ^18^F-PM-PBB3 PET scan (time frames: 18 × 10 s, 4 × 30 s, 5 × 1 min, 10 × 5 min, 3 × 10 min) was started for each animal simultaneously after receiving an IV injection of the radiotracer with an average dose of 19.3 ± 1.9 MBq via the tail vein.

All PET images were processed and analyzed by using the PMOD image analysis software (v3.5, PMOD Technologies Ltd., Zurich, Switzerland). To lower the possible bias from manual VOIs (volume of interests) drawing, an early phase PET template was created with the first 10 min mean PET image acquisition from each animal based on the rigid matching function of to the PMOD built-in mouse T2 MRI template (the procedure for the PET image template creation is shown in Appendix A) [40,41]. Then all other PET images were spatially normalized to this early phase PET template and the different VOIs, including the hippocampus, striatum, cortex, midbrain, and cerebellum from the MRI template were applied for image quantitation (the representative VOI selection image is shown in Figure 7). To avoid the difficulty of arterial blood sampling in the small animal study, a reference Logan graphical analysis (LGA) method was used. DVR values were determined at the VOI level using the cerebellum as reference tissue devoid of tau fibrils [34]. The equation used for the analysis is:(1)$$\frac{\int_{0}^{T}C_{voi}\left(t\right)dt}{C_{voi}\left(T\right)}=DVR\;\frac{\int_{0}^{T}C_{ref}\left(t\right)dt+C_{ref}\left(T\right)/{\left(k_{2}\right)}_{ref}}{C_{voi}\left(T\right)}+y_{intercept}$$
where C_voi_ and C_ref_ are the time–activity curves in the VOI and the reference region, respectively. The reference tissue efflux constant (k_2_)_ref_ is estimated from simplified reference tissue model. The slope of the linear portion of this Logan plot is thus the DVR [33,42,43]. SUVRs derived from stepwise time windows were regressed onto LGA to access the quantitative accuracy of SUVR and optimal scanning time window.

### 4.4. Ex Vivo Autoradiography

To further evaluate the tracer deposition in the transgenic mouse animal model, the 6-mo-old animals were sacrificed immediately after in vivo PET imaging procedure and applied for the ex vivo autoradiography as in our previous work [41]. Briefly, the animal was sacrificed with cervical dislocation and then the brain was quickly removed and frozen in dry ice. After embedding and frozen with the embedding buffer (OCT, tissue-tek), the brain tissue was sectioned with a thickness of 20 μm in sagittal orientation and collected on super frost slides. The tissue sections were dried up with a fan, then they were exposed to a phosphor imaging plate for at least 20 h. The plate was read with a FLA-5000 phosphor imaging reader with the PMT power setup of 650 V. The ex vivo autoradiogram was analyzed with the Multi Gauge v3.0 software (Fujifilm, Tokyo, Japan) to acquire the tracer uptake in different brain regions.

### 4.5. Immunofluorescence Analysis 

To confirm the overall expression of human tau protein in the rTg4510 mouse brains, 20-µm sagittal brain sections of the 6-mo-old animal group were applied for abnormal tau protein immunofluorescence staining. The antibody staining procedure was conducted by washing each slice with phosphate buffered saline (PBS) for 10 min 3 times at room temperature (RT). Each slice was then incubated in 0.3% Triton-X100 (Thermo Fisher Scientific, Waltham, MA, USA) for 1 h at RT. A non-specific blocking step was performed in blocking buffer (10% normal goat serum/5% bovine serum albumin (BSA) in PBS), followed by incubation in 20 μg/mL of Fab fragment goat anti-mouse IgG PBS solution (Jackson ImmunoResearch, West Grove, PA, USA) for 1 h at RT. The slice was then washed by PBS again, and incubated with anti-tau antibodies HT7 (1:1000 dilution) or AT8 (1:1000 dilution) overnight at 4 °C to detect total human tau and aggregated tau, respectively. On the following day, the slice was washed with PBS first, and then incubated with the secondary antibody (Jackson ImmunoResearch) for 1 h at RT. The slice was then transferred to a new tube, incubated with DAPI solution (1:1000 dilution) for 10 min at RT, and then washed with PBS. Each slice was then sealed in a glass slide with a coverslip with mounting medium (Thermo).

Consecutive overlapping images were taken by the ImageXpress Micro Confocal system (Molecular Devices, San Jose, CA, USA) using a 10× or a 20× objective lens for the antibody-stained slices. Post-hoc stitching functionality was applied to obtain the overall view of the sagittal section.

### 4.6. Statistical Analysis

Intergroup comparisons of regional SUVR values from ^18^F-PM-PBB3 mouse images were performed with the nonparametric one-way ANOVA test and Tukey’s multiple comparisons. All statistical calculations were performed by using the GraphPad Prism software, v5.0 (GraphPad Inc., San Diego, CA, USA).

## 5. Conclusions

In summary, this novel tau tracer, ^18^F-PM-PBB3, exhibited promising dynamic imaging ability to reach the plateau of the tracer uptake in the specific tau pathology deposition regions. Also, the 30-min optimal imaging time window of 40–70min post-injection was successfully applied for the in vivo cross-sectional PET study to differentiate the age-dependent tau pathology burden in the rTg4510 transgenic mouse model. With all the data presented herein, we strongly suggest that this tau imaging platform provides an opportunity to monitor the longitudinal tau pathology progression and to speed up new therapy development for tauopathy-related diseases.

## Figures and Tables

**Figure 1 molecules-25-01750-f001:**
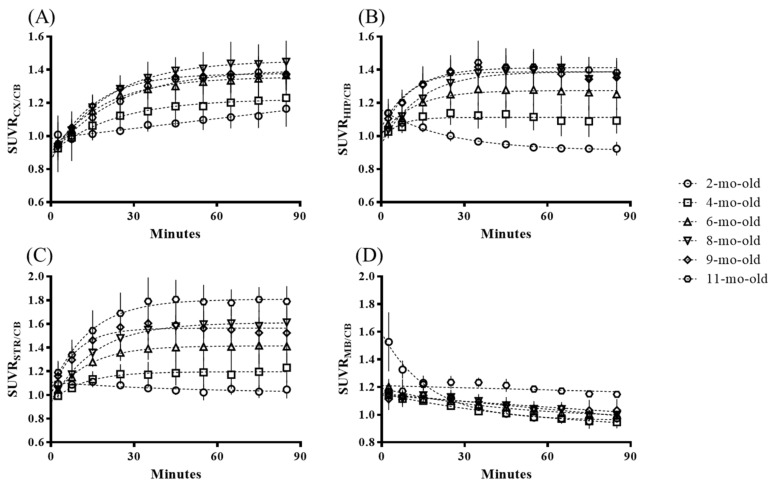
Mean regional standardized uptake value ratios (SUVR) time–activity curve of (**A**) cortex, (**B**) hippocampus, and (**C**) striatum, and (**D**) midbrain for each age group in the dynamic PET imaging acquisition studies. CX: cortex, HIP: hippocampus, MB: midbrain, STR: striatum, CB: cerebellum.

**Figure 2 molecules-25-01750-f002:**
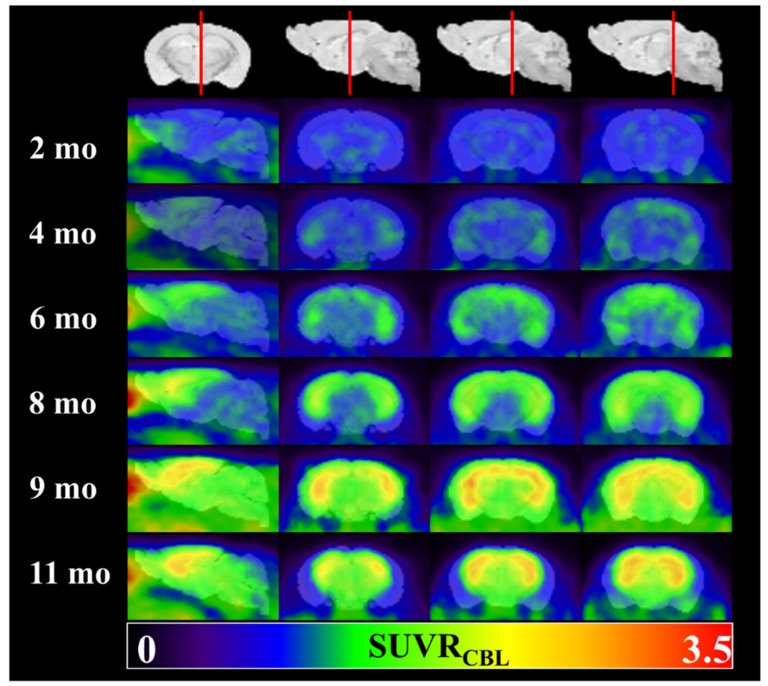
Representative ^18^F-PM-PBB3 PET images of different animal groups.

**Figure 3 molecules-25-01750-f003:**
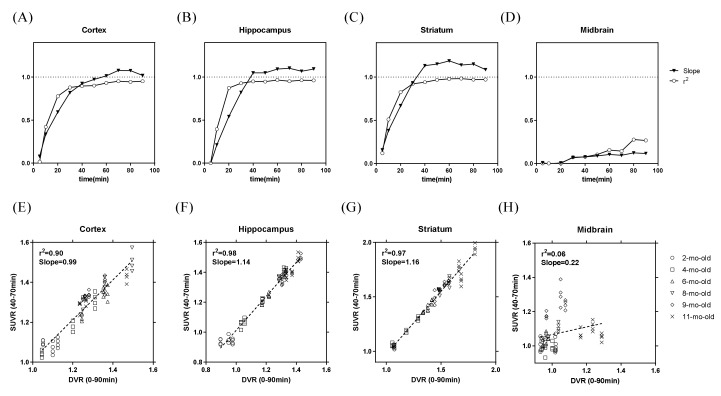
The r^2^ value and slope of the regression between the distribution volume ratios (DVR) and the standardized uptake value ratio (SUVR) measured at various time intervals in (**A**) cortex, (**B**) hippocampus, (**C**) striatum, and (**D**) midbrain. Correlation between DVR to SUVR at the time interval between 40–70 min post-injection for all age groups in (**E**) cortex, (**F**) hippocampus, (**G**) striatum, and (**H**) midbrain.

**Figure 4 molecules-25-01750-f004:**
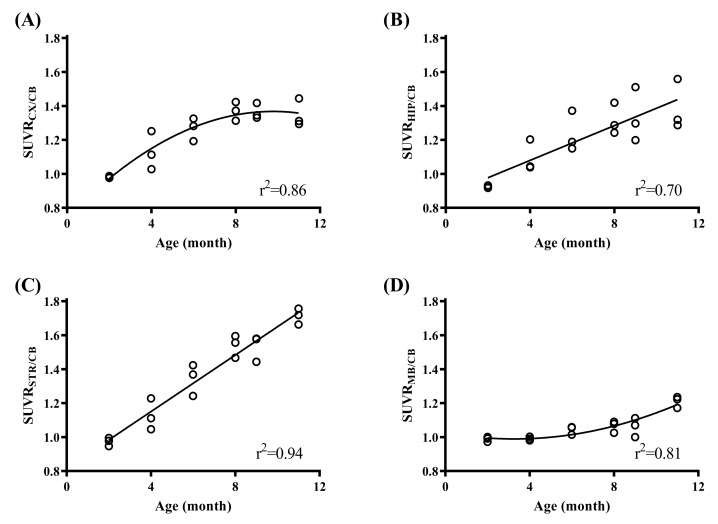
The scatter plots of regional ^18^F-PM-PBB3 standardized uptake value ratio (SUVR) across all age groups for regions of (**A**) cortex, (**B**) hippocampus, (**C**) striatum, and (**D**) midbrain using cerebellum as the reference region. CX: cortex, HIP: hippocampus, MB: midbrain, STR: striatum, CB: cerebellum.

**Figure 5 molecules-25-01750-f005:**
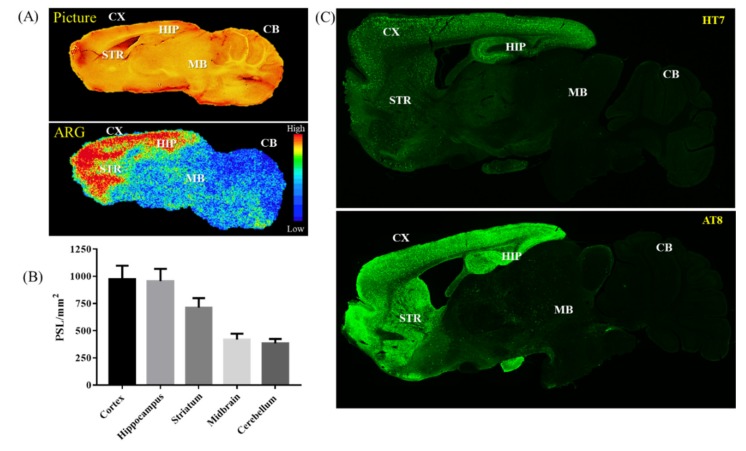
Correlations of ^18^F-PM-PBB3 radioactivity distribution with tau immunohistochemical study in 6-mo-old rTg5410 mouse brain sections. (**A**) Representative photograph of the sagittal brain section and its autoradiography. (**B**) Quantitative measurement of radioactivity in different brain regions of the rTg4510 mouse. (**C**) Neocortical/hippocampal human tau (HT7 antibody) and hyperphosphorylated tau (antibody AT8) immunofluorescent on sagittal brain sections at the level adjacent to the autoradiography study. (CX: cortex, STR: striatum, HIP: hippocampus, MB: midbrain, CB: cerebellum.).

**Figure 6 molecules-25-01750-f006:**
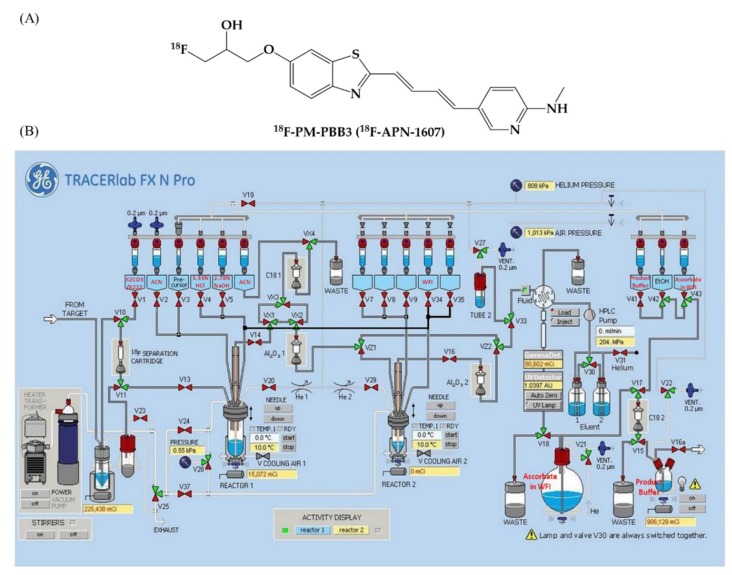
The chemical structure of ^18^F-PM-PBB3 (^18^F-APN-1607) (**A**) and its automatic radiosynthesis procedure scheme (**B**).

**Figure 7 molecules-25-01750-f007:**
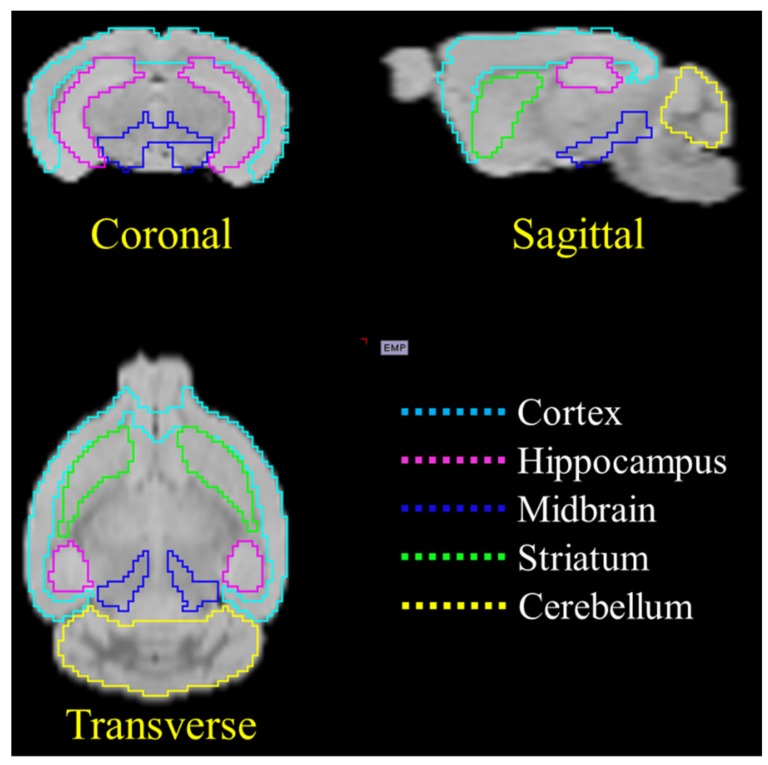
Representative VOIs from the T2 template MR image used for PET image quantification analysis.

**Table 1 molecules-25-01750-t001:** Quantification ^18^F-PM-PBB3 standardized uptake value ratio (SUVR) values for all age groups.

	2-mo-old			4-mo-old			6-mo-old			8-mo-old			9-mo-old			11-mo-old		
**CX**	**0.98**	±	**0.01**	**1.13**	±	**0.11**	**1.27**	±	**0.07***	**1.37**	±	0.06**	1.36	±	0.05**	1.35	±	0.08*
HIP	0.92	±	0.01	1.10	±	0.09	1.24	±	0.12*	1.32	±	0.09*	1.34	±	0.16*	1.39	±	0.15*
STR	0.97	±	0.02	1.13	±	0.09	1.34	±	0.09*	1.54	±	0.07**	1.53	±	0.08**	1.71	±	0.05***
MB	0.99	±	0.02	0.99	±	0.01	1.04	±	0.03	1.06	±	0.03	1.06	±	0.06	1.21	±	0.03**
* *p* < 0.05; ** *p* < 0.01; *** *p* < 0.001																		

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
