# Peer review of "Characterization of ^18^F-PM-PBB3 (^18^F-APN-1607) Uptake in the rTg4510 Mouse Model of Tauopathy"

_molecules, 2020, doi:10.3390/molecules25071750_

Round 1

Reviewer 1 Report

The manuscript provided by Weng et al. deals with the in vivo preclinical evaluation of the newly available tau radiotracer 18F-PM-PBB3 to determine an effective time window for imaging. Therefore, they use the rTg4510 animal model with 90 min PET scanning in a preclinical PET/CT. Following VOI analysis SUVR (cerebellum as reference region) and DVU were calculated following kinetic modeling with PMOD software tool. Resultantly, a time window from 40-70 min p.i. is suggested solely based on the correlation of the both aforementioned parameters. Although the methodology is sound and well performed more information (results from a second method) should be added to justify the validity of this correlation analysis for all animal age groups. In particular, its not clear to the reader why this correlation points to the perfect scanning window. This needs to be better explained. Furthermore, how are these results translatable to human entity with regard of the adaption of time scale? 40-70 min in mice will equal some hours in humans. Can allometric scaling be applied and help to estimate from animal to human time scale? I would kindly ask the authors to perform a major revision of this particular parts. Furthermore, English language needs intensive revision as well as text editing (missing spaces, isotope lower script in supplement etc.). I have marked some critical points throughout the manuscript. Following the addressed corrections this very important manuscript in the field of tau tracer development and application will be significantly improved and fit to the journal scope/quality. Specific comments to be addressed:

Line 21-22: Missing/additional spaces between “ages.Different” and “n =3”

Line 33: Missing space

Line 40-41: Repetition of “The optimal imaging time window of 40-70 min post-injection for the novel tau PET tracer” please delete

Line 49: missing space

Line 88: The radiotracer PM-PBB3 is investigated in a clinical study: NCT03625128. Please provide more background information as well as results from this study. Why is this preclinical study performed and necessary if the tracer is already in clinical study phase? Again, is the time window applicable to human entity? Please be more specific on the aim of this preclinical investigation!

Line 90: “comparable binding characteristics to 11C-PBB3 in vitro.” Please provide reference.

Line 96: missing space

Line 105-110: This sentence needs to be rewritten. It’s absolutely not clear to me what the authors like to explain.

Line 106: What is meant by “clear pictures”. Please use better and scientific wording. Is there a measure for “clear”?

Line 106: “in the all brain regions” revise

Line 107: “the signal of the tracer began to accumulate”. Who is accumulating, the tracer or the signal? Please revise.

Line 107: “afterward”. At later time points? Please be more specific here. What information should be transferred by this sentence.

Line 109: “compared to the older animals”. From Figure 2 it seems like the older animals show higher uptake in ctx and hpc. Hence, should the sentence read “compared to younger animals” instead?

Line 109: Regarding brain atrophy: How is this visible in the provided data? Please clarify and describe in more detail.

Figure 2: Please make the SUV only data available in the supplemental data as well. MDPI encourage to make all data available, please follow these guidelines.

Line 123: “..after a scanning time of 40 min..” The data before 40 min is not provided here and so its not possible to follow this statement. Please make this data available as well and include into the main manuscript as it helps to understand what is written in the text.

Line 128: “R2”--> r2?

Line 132: space missing

Line 152: delete “can” and “was”

Line 153: delete “was”

Line 158: Why was 6-mo old chosen while 9-mo old shows even higher accumulation. Please explain.

Line 159-160: Sentence seems incomplete

Figure 5: It would be very interesting to see how midbrain behaves in the oldest group, to confirm the theory of late effect to midbrain. Please consider to add immunofluorescence data for all animal groups!

Figure 5: Caption incomplete. Please describe and mention A and B part of the image as well.

Line 182: delete “can easily”

Line 190: Please provide binding affinities for PM-PBB3 as well throughout the manuscript and discuss.

Line 192: Please discuss: Why is DVR used as a "gold standard" here? Please justify within the manuscript. Please also describe DVR method in more detail in methods part.

Line 194: “….for other tauopathy research” Please specify.

Line 197: “best correlation with the DVR values.” What is the background of using DVR and what is the consequence out of this result? Please discuss.

Line 201: In this regard as mentioned before, please discuss the feasibility of translation of results to human entity. Allometric scaling to human? Are results translatable to human research or other animal models?

Line 210: “…age of 7-mo”. 7 vs. 11 month (in particular when using mouse models) is in my opinion not a comparable result! Its a significant deviation and needs to be discussed here!

Line 213: “PET in vivo resolving power issue..” With this regard, please consider further ex vivo methods to proof tau pathology among all ages. At least immunofluorescence and autoradiography should be used for the other age groups as well! Are brains still frozen and available?

Line 228: “GE healthy care” --> GE health care?

Line 231: How can this radiochemical purity be explained? Please provide starting activity in the main manuscript.

Line 244: Please provide information about the scanner in the main manuscript as well and not only the supplement.

Line 248: Missing space

Line 249: “immediately” Simultaneous injection and PET start or with time delay?

Line 255: “T2 template” Is it the one included in PMOD or another one? Please specify.

Line 260: “cerebellar VOI” Please explain if and why this is a suitable reference region with regard to specific and non-specific binding. Please describe also about radio metabolites, crossing BBB? Metabolite correction needed?

Line 273: add something like “…and collected on super frost slides.”

Line 283: missing space after 20

Line 292: missing space “theantibody-stained”

Author Response

Response to Reviewer 1 Comments

Point 1: Line 21-22: Missing/additional spaces between “ages.Different” and “n =3”
Response 1: Thanks for the comment, the abstract is now re-written to give concise and meaningful findings of the study. Accordingly, the missing/additional spaces are corrected.

Point 2: Line 33: missing space
Response 2: Thanks for the comment, the abstract is now re-written and the missing space is corrected.

Point 3: Line 40-41: Repetition of “The optimal imaging time window of 40-70 min post-injection for the novel tau PET tracer” please delete
Response 3: Thanks for the comment, the abstract is now re-written and the repetition is removed.

Point 4: Line 49: missing space
Response 4: Add missing space in “world[1]” as suggested (line 39).

Point 5: Line 88: The radiotracer PM-PBB3 is investigated in a clinical study: NCT03625128. Please provide more background information as well as results from this study. Why is this preclinical study performed and necessary if the tracer is already in clinical study phase? Again, is the time window applicable to human entity? Please be more specific on the aim of this preclinical investigation!
Response 5: Thanks for the important comment. Indeed, the radiotracer has been recently investigated in a clinical study: NCT03625128. Both optimal scanning time window and image pattern in AD and non-AD human subjects were studied and presented in the Human Amyloid Imaging conference at Miami, FL, USA in 2019. The full manuscript has been submitted and under review process. This promising result has raised the interest of translational drug development using this tracer as a biomarker in preclinical animal research. Of note, the time window and image uptake pattern elucidated in this preclinical animal study may not be directly applicable to the human entity, vice versa. In this regard, the independent information illustrated from our work would be important for understanding the ability to detect tau protein aggregation in the tauopathy animal model. The result may provide valuable information for design animal research protocols in anti-tau therapeutic screening and drug development study in the
future. And this is the specific aim of the preclinical investigation. As suggested, we now re-written the abstract, add the following paragraph in the introduction (line 81-86 in the revised manuscript):

" Recently, Huang et al investigated the optimal scanning time window of this novel tracer [27] and showed promising results for detecting tau distribution in both AD and non-AD subjects in a phase 0 study (manuscript under review). Currently, the strategy for developing a new treatment for neurodegenerative disorder relies partially on preclinical animal models. We propose the ability of non-invasive monitor tau aggregation using this novel radiotracer for addressing the bottleneck in drug discovery and development for dementia."

Point 6: Line 90: “comparable binding characteristics to 11C-PBB3 in vitro.” Please provide reference.
Response 6: Thanks for the comment. Relevant binding characteristics and two references are added to the introduction (line 79-80).
“…comparable binding characteristics to 11C-PBB3 in vitro (Kd = 9.9 nM for 18F-PMPBB3 and Kd = 6.3 nM for 11C-PBB3 on the AD frontal cortex) [24, 26].”
24. Ono, M., et al., Distinct binding of PET ligands PBB3 and AV‐1451 to tau fibril strains in neurodegenerative tauopathies. Brain, 2017. 140(3): p. 764‐780.
26. Ono, M., et al., Development of novel tau PET tracers, [18F]AMPBB3 and [18F]PM‐PBB3. Presented at 11th Human Amyloid Imaging, Miami, FL, USA, Jan 2017. P34. Available online: https://hai.worldeventsforum.com/past_editions/ (accessed on 31 March 2020).

Point 7: Line 96: missing space
Response 7: Thanks for the comment. The missing space in “1.5hr” is now corrected (line 92).

Point 8: Line 105-110: This sentence needs to be rewritten. It’s absolutely not clear to me what the authors like to explain.
Response 8: Thank for the comment. We agree that the original Figure 2 result may confuse, and the paragraph is now re-written as follows (line 99-108).
“Figure 2 shows representative rainbow color-coded animal PET images at different ages overlaid on a T2 MR template. The static PET images were summed from dynamic frames obtained between 40-70min post tracer injection. Sagittal plane and three coronal views in hippocampus regions were illustrated for comparison. 18F-PM-PBB3 uptake in SUVR showed no significant difference in the cortical region of 4-mo-old transgenic mice when compared to that of the 2-mo-old baseline group (+15.0%, not significant). The radioactivity uptake was elevated in 6-mo-old animal (+28.9%, p<0.05), and increased further in 9-mo-old group (+38.8%, p<0.01). The trend of
increased SUVR value becomes steady or slightly reduced in 11-mo-old animal (+37.3%, p<0.05). Of note, when compared to the overlaid MR template, the brain PET size was smaller at this age group as compared to those of the younger mice, suggesting cortical atrophy in the elder animal”

Point 9: Line 106: What is meant by “clear pictures”. Please use better and scientific wording. Is there a measure for “clear”?
Response 9: Thanks for the comment. According to point 8, the paragraph is now rewritten. The “clear picture” is now replaced with a scientific wording: “18F-PM-PBB3 uptake in SUVR showed no significant difference in the cortical region of 4-mo-old transgenic mice when compared to that of the 2-mo-old baseline (+15.3%, not significant).”. (line 102-106)

Point 10: Line 106: “in the all brain regions” revise
Response 10: Thanks for the comment. The paragraph is now rewritten as indicated in point 8. The “in the all brain regions” is now replaced with “cortical region”. (line 103)

Point 11: Line 107: “the signal of the tracer began to accumulate”. Who is accumulating, the tracer or the signal? Please revise.
Response 11: Thanks for the comment. The paragraph is now rewritten as indicated in point 8. The “accumulate” is now replaced with “The radioactivity uptake was elevated in 6-mo-old animal (+28.9%, p<0.05), and increased further in 9-mo-old group (+38.8%, p<0.01).” (line 104-105).

Point 12. Line 107: “afterward”. At later time points? Please be more specific here. What information should be transferred by this sentence.
Response 12: Thanks for the comment. The paragraph is now rewritten as indicated in point 8. The confusing word “afterward” is now replaced with “increased further” (line 104-105).

Point 13. Line 109: “compared to the older animals”. From Figure 2 it seems like the older animals show higher uptake in ctx and hpc. Hence, should the sentence read “compared to younger animals” instead?
Response 13. Thanks for the comment. The original sentences were not clear and misleading, we now rewrote the paragraph as indicated in point 8. The sentence is now replaced with “The radioactivity uptake was elevated in 6-mo-old animal (+28.9%, p<0.05), and increased further in 9-mo-old group (+38.8%, p<0.01). The trend of increased SUVR value becomes steady or slightly reduced in 11-mo-old animal (+37.3%, p<0.05).” (line 104-106).

Point 14. Line 109: Regarding brain atrophy: How is this visible in the provided data? Please clarify and describe in more detail.
Response 14: Thanks for the comment. The original sentences were not clear and misleading, we now rewrote the paragraph as indicated in point 8. New sentences are added to the result as follows “Of note, when compared to the overlaid MR template, the brain PET size was smaller at this age group as compared to those of the younger mice, suggesting cortical atrophy in the elder animal.” (line 106-108). A detail discussion is also added to lines 224-232.

Point 15. Figure 2: Please make the SUV only data available in the supplemental data as well. MDPI encourage to make all data available, please follow these guidelines.
Response 15: Thank for the comment. The figure of the SUV only data is now added in the revised supplemental data (Figure S3).

Point 16. Line 123: “..after a scanning time of 40 min..” The data before 40 min is not provided here and so its not possible to follow this statement. Please make this data available as well and include into the main manuscript as it helps to understand what is written in the text.
Response 16: Thank for the comment. We now added the data before 40 min as suggested. The r2 value and slope of the regression between the DVR and SUVR measured at various time intervals in (A) cortex, (B) hippocampus, (C) striatum, and (D) midbrain are now provided in the new figure 3 (line 125-129).

Point 17. Line 128: “R2”--> r2?
Response 17: Thank for the comment. The typo “R2” is now corrected to “r2” (line 126).

Point 18. Line 132: space missing
Response 18: Thank for the comment. The missing space is now corrected (line 131).

Point 19. Line 152: delete “can” and “was”
Response 19: Thank for the comment. We agree that the original sentence was not clear and misleading, we now rewrote as follows (line 149-151).
“The high-resolution autoradiography revealed a spatial distribution of radioactivity in the cortex, striatum, and hippocampus comparable to those in PET image findings (Figure 2).”

Point 20. Line 153: delete “was”
Response 20: Thank for the comment. We agree that the original sentence was not clear and misleading, we now rewrote as indicated in point 20 (line 149-151).

Point 21. Line 158: Why was 6-mo old chosen while 9-mo old shows even higher accumulation. Please explain.
Response 21: Thanks for the comment. According to our PET quantification result, 6-mo-old is the earliest age group of transgenic mice that showed significant 18F-PMPBB3 uptake in the brain cortex (table 1). In this regard, it is important to evaluate whether the radioactivity uptake correlates to the Tau signal as illustrated by immunohistochemistry study. This information would be important for a proper therapeutic drug design using transgenic mice and 18F-PM-PBB3 PET as a biomarker. Accordingly, the explanation is now added as follows (line 146-147).
“To confirm the 18F-PM-PBB3 uptake from animal PET in the earliest age group of the rTg4510 transgenic mice (see Table 1), ex vivo autoradiography study was performed in a 6-mo-old animal.”

Point 22. Line 159-160: Sentence seems incomplete
Response 22: Thanks for the correction. We agree that the original sentence was not clear and misleading, we now rewrote the paragraph as follows (line 157-161).
“To establish whether radioactivity distribution co-localized to the brain regions express transgenic P301S-htau, adjacent brain sections from the above autoradiography study were immunolabeled for the presence of human tau (HT7 antibody) and hyperphosphorylated tau (antibody AT8). Immunofluorescent imaging (figure 5C) showed prominent signals in the cortex, hippocampus, and striatum comparable to the autoradiography findings (Figure 5A).”

Point 23. Figure 5: It would be very interesting to see how midbrain behaves in the oldest group, to confirm the theory of late effect to midbrain. Please consider to add immunofluorescence data for all animal groups!
Response 23: Thank you very much for the important point, We appreciate the Reviewer’s very constructive suggestion to make this manuscript more powerful. Unfortunately, we couldn’t make it here right now because of the brain tissues are not available. And we will include this experiment in our next project.

Point 24. Figure 5: Caption incomplete. Please describe and mention A and B part of the image as well.
Response 24: Thanks for the correction. The figure legend is now revised as follows (line 167-173).
“Figure 5. Correlations of 18F-PM-PBB3 radioactivity distribution with tau immunohistochemical study in 6-mo-old rTg5410 mouse brain sections. (A) Representative photograph of the sagittal brain section and its autoradiography. (B) Quantitative measurement of radioactivity in different brain regions of the rTg4510
mouse. (C) Neocortical/hippocampal human tau (HT7 antibody) and hyperphosphorylated tau (antibody AT8) immunofluorescent on sagittal brain sections at the level adjacent to the autoradiography study. (CX: cortex, STR: striatum, HIP: hippocampus, MB: midbrain, CB: cerebellum.)”

Point 25. Line 182: delete “can easily”
Response 25: Thanks for the comment. The sentence is now revised. (line 186).

Point 26. Line 190: Please provide binding affinities for PM-PBB3 as well throughout the manuscript and discuss.
Response 26: Thanks for the comment. We now provide the binding affinities information in the introduction (line79-80)
“… shown comparable binding characteristics to 11C-PBB3 in vitro (Kd = 9.9 nM for 18F-PM-PBB3 and Kd = 6.3 nM for 11C-PBB3 on the AD frontal cortex) [24, 26].”
We also modified the discussion as suggested (line 192-198).
“However, 18F-THK523 retention was significantly lower in gray matter than in white matter, suggesting insufficient binding affinity of this tracer to tau aggregates [32]. While 18F-PM-PBB3 washout fast from cerebellum reference region (Figure 1) makes it feasible for visual inspection of tau deposits in the brain cortex (Figure 2). Owing to
its high binding affinity [26], the radioactivity uptake measured by SUVR may potentially differentiate stage of tauopathy in each age group of the Tg4510 animals to facilitate translational research.”

Point 27: Line 192: Please discuss: Why is DVR used as a "gold standard" here? Please justify within the manuscript. Please also describe DVR method in more detail in methods part.
Response 27: Thanks for the comment. We now add a paragraph to discuss why DVR sued as a “gold standard” here. (line 199-206)
“To relate the radioactivity uptake measured by PET to the underlying biochemical process, the utilization of mathematical models to describe tracer kinetics is necessary.These methods are generally derived based on the theory for conventional tracer kinetic models such as spectral analysis, compartmental models, and graphical methods in which the plasma input function may be replaced by a reference region devoid of the specific binding sites [33]. Thus, the DVR measured by Logan plot with cerebellum as a reference [34] was used as a “gold standard” in our study. To further reduce the complexity of the PET study, estimate the transient equilibrium time window when SUVR as a proxy for DVR values was established”
We also describe DVR method in more details as follows (line 286-295).
“DVR values were determined at the VOI level using the cerebellum as reference tissue devoid of tau fibrils [34]. For the formula used, please see the attached file.

Point 28. Line 194: “….for other tauopathy research” Please specify.
Response 28: Thanks for the comment. Examples of other tauopathy such as progressive supranuclear palsy and cortical basal degeneration are now added (line 208-209).

Point 29. Line 197: “best correlation with the DVR values.” What is the background of using DVR and what is the consequence out of this result? Please discuss.
Response 29: Thanks for the comment. The background of using DVR as the gold standard for quantification is now added to the discussion section as indicated in response 27 (line 199-206) SUVR value measured out of this time window showed less accuracy in estimate the radiotracer uptake, and the r2 value and slope of the regression between the DVR and SUVR measured at all time intervals are now added to Figure 3 (A-D) for discussion.
(line 125-129)

Point 30. Line 201: In this regard as mentioned before, please discuss the feasibility of translation of results to human entity. Allometric scaling to human? Are results translatable to human research or other animal models?
Response 30: Thanks for the comment. It is of interest to compare tracer characteristics when used in humans and animals. Often time, there is no easy way to add a scaling factor to transform human data in animals or vice versa. The human optimal scanning time window result is now added for comparison. The time window or image uptake pattern elucidated in our preclinical animal study may not directly applicable to the
human entity, however, this information would be important for understanding the ability to detect tau protein aggregation in tauopathy animal model. The result of our study provides valuable information for design animal research protocols in anit-tau therapeutic screening and drug development study in the future. The following
discussion is now added to the revised manuscript as suggested. (line 216-219)
“This difference even larger when compared to a human study showing 90-110min post-injection as an optimal scanning time window [27]. Since there is no simple allometric scaling factor to transform animal results into human, independent study design for each species and disease model is necessary for its application”

Point 31: Line 210: “…age of 7-mo”. 7 vs. 11 month (in particular when using mouse models) is in my opinion not a comparable result! Its a significant deviation and needs to be discussed here!
Response 31: Thank you for point out this important comment. Indeed, 7-mo vs 11-mo is not a comparable result. The 18F-PM-PBB3 uptake pattern behaves differently in various brain regions. The cortical brain region reached its plateau at the age of around 8 mo-old, which is comparable to the C-11 PBPB3 result. Inversely, the uptake in
subcortical regions such as hippocampus and striatum increases continuously till 11- mo-old is not comparable to the C-11 PBB3 result. A discussion for the above deviation is revised (lines 221-236)
Also, the fitting curves for Figure 4A and 4D are revised (line 142).

Point 32. Line 213: “PET in vivo resolving power issue..” With this regard, please consider further ex vivo methods to proof tau pathology among all ages. At least immunofluorescence and autoradiography should be used for the other age groups as well! Are brains still frozen and available?
Response: We appreciate the Reviewer’s very constructive suggestion to make this manuscript more powerful. Unfortunately, we couldn’t make it here right now because of the brain tissues are not available. And we will include this experiment in our next project.

Point 33. Line 228: “GE healthy care” --> GE health care?
Response 33: The word is now corrected. (line 258).

Point 34. Line 231: How can this radiochemical purity be explained? Please provide starting activity in the main manuscript.
Response 34: Thanks for the comment. The radiochemical purity of 18F-PM-PBB3 was determined using an HPLC system, and the starting activity was 72.72±10.5GBq. (line 259-262).
Additional detail regarding HPLC analysis is also added to the supplementary material (line 40-52)

Point 35. Line 244: Please provide information about the scanner in the main manuscript as well and not only the supplement.
Response 35: Thanks for the comment. The information is now added to the method section (line 278-279)

Point 36. Line 248: Missing space
Response 36: Thanks for the comment. Space is now added to the method section (line 282).

Point 37. Line 249: “immediately” Simultaneous injection and PET start or with time delay?
Response: Thanks for the comment. The “immediately” means the PET acquisition was started simultaneously after injection without a time delay (line 283).

Point 38. Line 255: “T2 template” Is it the one included in PMOD or another one? Please specify.
Response 38: Thanks for the comment. “PMOD built-in mouse T2 MRI template “ is now added to the method section (line 289).

Point 39. Line 260: “cerebellar VOI” Please explain if and why this is a suitable reference region with regard to specific and non-specific binding. Please describe also about radio metabolites, crossing BBB? Metabolite correction needed?
Response 39: Thanks for the comment. The reference region should be devoid of the targeted binding site or pathology and should provide robust reference values for semiquantitative analyses of the tau pathology. Firstly, the cerebellum has been reported as devoid of tau (Frontiers in Aging Neuroscience. 2017;9). Secondly, our present
immunohistochemical result showed no tau deposition in the cerebellum of aged rTg4510 mice. Therefore, we applied the cerebellum as a reference region for the SUVR and DVR analysis. A statement was added to the method section (line 295). We did not investigate the radio metabolites in our study due to technical limitations on the mouse blood sampling. In brain studies, the radioactive metabolites do not usually pass the blood-brain barrier (BBB) due to more polar than the parent compound. According to the report from Ono M et al., they observed fewer radiometbolites of 18FPM-PBB3 in mouse plasma and brain than those of 11C-PBB3. Also, from our autoradiography result, there was no significant off-target binding signal, suggesting brain radiometabolite may be negligible in our study. Since radiometabolites data was not available in our study, arterial input function with metabolite corrected plasma data
was avoided and replaced by using a reference tissue model as “gold standard”. Reference tissue methods also avoid the errors in the determination of the fraction of parent tracer in plasma with HPLC. However, we will take a very serious consideration to include this experiment in our next project. A description concerning the radiometabolites limitation is added to the discussion (line 246-254).

Point 40: Line 273: add something like “…and collected on super frost slides.”
Response 40: Thanks for the comment. “…and collected on super frost slides.” was added to the method (line 314).

Point 41: Line 283: missing space after 20
Response 41: Thanks for the correction. The “20 g/ml” is now modified in the revised manuscript. (line 325)

Point 42. Line 292: missing space “theantibody-stained”
Response 42: Thanks for the correction. The “the antibody-stained” is now modified in the revised manuscript. (line 334)

Reviewer 2 Report

The work reported by Weng et al represents an interesting study on developing PET tracers for imaging tau progression. However, the points below need to be addressed before this work goes further:

  1. The abstract is lengthy. It needs to be concise and deliver the main findings of the study
  2. The authors cote reference 34 for the tracer and states that new tracer was done with some modifications. The reference is for C11 tracer and they report F18 tracer. So this is not the appropiate reference.
  3. The structure of the tracer is not revealed. The chemistry of the synthesis is not shown in the supplementary file.
  4. No HPLC traces are displayed to showcase the purity of tracer
  5. How did the authors confirm the identity of the tracer?  co-injection with cold standard? No details are provided.
  6. The stability of the tracer needs to be addressed. Both in vitro and in vivo stability

Author Response

Response to Reviewer 2 Comments

Point 1. The abstract is lengthy. It needs to be concise and deliver the main findings of the study
Response 1: Thanks for the comment. The abstract is now completely re-written to give concise and meaningful findings of the study in the revised manuscript as suggested. (line 17-35)

Point 2. The authors cote reference 34 for the tracer and states that new tracer was done with some modifications. The reference is for C11 tracer and they report F18 tracer. So this is not the appropiate reference.
Response 2: Thanks for the comment. We agree with the reviewer’s point, and reference 34 is now removed and the method of radiochemistry was modified in the manuscript accordingly (line 257-259).

Point 3. The structure of the tracer is not revealed. The chemistry of the synthesis is not shown in the supplementary file.
Response 3: Thanks for the comment. The chemical structure was previously described in the following paper: [Schöll M, Maass A, Mattsson N, et al. Biomarkers for tau pathology. Molecular and Cellular Neuroscience. 2019;97:18-33.], and is now added in Figure 6A as suggested (line 264). Additional radiosynthesis details, the method for radiochemical purity, and HPLC conditions and profiles are now provided in the revised supplementary file S1. for review (line S3-52).

Point 4. No HPLC traces are displayed to showcase the purity of tracer. How did the authors confirm the identity of the tracer? co-injection with cold standard? No details are provided.
Response 4: Thanks for the comment. The reviewer is correct about the study procedure. The radiochemical purity was determined by radio-HPLC. We confirmed the identity of the tracer with co-injection of a cold standard reference. The method section is now revised (line 259-262). A representative HPLC chromatography of this radiotracer and standard are now added to the supplementary materials as suggested (line S40-52).

Point 5. The stability of the tracer needs to be addressed. Both in vitro and in vivo stability
Response 5: Thanks for the comment. The stability of the tracer has been studied up to 6h after production. In addition to stability, the radiopharmaceutical was evaluated for pH, visual appearance, chemical, and radiochemical purity, after storage in the final dosage form for up to 6.7 h after the end of synthesis (EOS) at room temperature. The product remained within specifications at all time points. The expiration time was set at 6 h after the end of synthesis (EOS). The description of its result is now added to the method section as suggested. (line 260-262)
Regarding the in vivo stability, we appreciate the Reviewer’s very constructive suggestion to make this manuscript more powerful. Unfortunately, we did not investigate the radio metabolites in our study due to technical limitations on the mouse blood sampling. In brain studies, the radioactive metabolites do not usually pass the blood-brain barrier (BBB) due to more polar than the parent compound. According to the report from Ono M et al., they observed fewer radiometbolites of 18F-PM-PBB3 in mouse plasma and brain than those of 11C-PBB3. Also, from our autoradiography result, there was no significant off-target binding signal, suggesting brain radiometabolite may be negligible in our study. Since radiometabolites data was not available in our study, arterial input function with metabolite corrected plasma data was avoided and replaced by using a reference tissue model as “gold standard”. Reference tissue methods also avoid the errors in the determination of the fraction of parent tracer in plasma with HPLC. However, we will take a very serious consideration to include this experiment in our next project. A description concerning the radiometabolites limitation is added to the discussion (line 246-254)

Round 2

Reviewer 1 Report

Dear authors,

thanks for answering all my questions into detail and made changes. No further comments from my side. I am looking forward to see this nice manuscript published soon!

Reviewer 2 Report

The authors have addressed all my concerns